# High-fat diet disturbs lipid raft/TGF-β signaling-mediated maintenance of hematopoietic stem cells in mouse bone marrow

François Hermetet[1,2], Anne Buffière[1,2], Aziza Aznague[1,2], Jean-Paul Pais de Barros[2,3], Jean-Noël Bastie[1,2,4], Laurent Delva[1,2] & Ronan Quéré [1,2]

Despite recent in vivo data demonstrating that high-fat diet (HFD)-induced obesity leads to major perturbations in murine hematopoietic stem cells (HSC), the direct role of a HFD is not yet completely understood. Here, we investigate the direct impact of a short-term HFD on HSC and hematopoiesis in C57BL/6J mice compared with standard diet-fed mice. We detect a loss of half of the most primitive HSC in the bone marrow (BM) cells of HFD-fed mice, which exhibit lower hematopoietic reconstitution potential after transplantation. Impaired maintenance of HSC is due to reduced dormancy after HFD feeding. We discover that a HFD disrupts the TGF-β receptor within lipid rafts, associated to impaired Smad2/3-dependent TGF-β signaling, as the main molecular mechanism of action. Finally, injecting HFD-fed mice with recombinant TGF-β1 avoids the loss of HSC and alteration of the BM's ability to recover, underscoring the fact that a HFD affects TGF-β signaling on HSC.

[1] Signaling and Physiology in Hematological Research, INSERM, UMR1231, Université Bourgogne Franche-Comté, Dijon 21000, France. [2] LipSTIC LabEx, Fondation de Coopération Scientifique Bourgogne Franche-Comté, Dijon 21000, France. [3] Plateforme de Lipidomique, Université Bourgogne Franche-Comté, Dijon 21000, France. [4] Hôpital Universitaire François Mitterrand, Service d'Hématologie Clinique, Dijon 21000, France. Correspondence and requests for materials should be addressed to R.Qéré. (email: ronan.quere@inserm.fr)

In the hematopoietic system, hematopoietic stem cells (HSC) reside at the top of the hematopoietic hierarchy and have capacities of self-renewal and differentiation which are essential for the lifelong sustenance of the stem cell pool and the production of all types of blood cells, respectively[1,2]. Both of these abilities are finely regulated by both cell-intrinsic and cell-extrinsic mechanisms involving cytokines, transcription factors and cell–cell contacts[3], as well as epigenetic regulation[4]. More recently, several metabolic pathways have been recognized as regulatory elements of HSC self-renewal, commitment, and specification to the different lineages[5]. Concerning bioenergetic signaling in HSC, glucose and amino-acid-mediated metabolic networks are now well known for regulating HSC potential[6–8], while the lipid-dependent regulation of HSC remains unidentified. Although mammalian regenerative tissues[9–11], including hematopoietic tissue[12–14], are known to respond to dietary signals, little is known about how high-fat diets (HFD), collectively known as pro-obesity or Western diets, regulate tissue stem/progenitor cell function. Some recent studies on wild type rodent models have shown that HFD-induced obesity triggers significant perturbations of HSC and homeostasis of the hematopoietic system[14–18], but it is difficult to ascertain whether these alterations are the result of a direct effect such as changes in lipid metabolism in HSC, or only related to the pathophysiology of obesity, inflammation or diabetes. Fatty acid metabolism supports both the biosynthetic and bioenergetic requirements of cell proliferation and survival while lipids are essential components of plasma and organelle membranes. Lipid rafts (LR) are cholesterol-enriched patches located in the plasma membrane, and the dynamic protein assembly in these LR can be modified by a disturbance in the lipid composition of cells[19]. As platforms for membrane trafficking and signal transduction, LR are master regulators of cytokine function, cell cycle activity and are also involved in the retention/dormancy of HSC in bone marrow (BM)[20–22].

In this study on mice, we found that ingesting a HFD for as little as 4 weeks can affect the organization of LR on the surface of HSC, which in turn disturbs the LR/TGF-β signaling-mediated quiescence of HSC and affects their maintenance in mouse BM. Here, we build upon the growing body of literature implicating dietary and metabolic control as important regulators of stem cell populations with a special focus on hematopoietic tissue.

## Results

**HSC expresses high level of lipid rafts.** We stained various hematopoietic cell populations with the cholera toxin subunit B that binds to the ganglioside GM1 (one of the main components of LR). We then observed that HSC had high level of LR, but the levels decreased in more mature progenitor cells (99.1% for lineage negative (Lin⁻) Sca1⁺ c-Kit⁺ (LSK) CD48⁻ CD150⁺ (SLAM) and 36.6% for the Lin⁻ cells) (Fig. 1a). We identified two distinct populations of LSK-CD34⁻ cells: half displayed high levels of LR (LR^hi), while the other half had low detectable levels of LR (LR^lo) (Fig. 1b). LR^hi cells were enriched with the most primitive HSC (SLAM; 46% versus 2% for LR^lo) (Fig. 1c). When we characterized the propensities of both types of cells to engraft in lethally irradiated recipient mice, only LR^hi cells (among LSK-CD34⁻ cells) showed a marked ability to reconstitute 16 weeks after the transplantation (Fig. 1d), meaning that this population was enriched in long-term reconstituting HSC.

**HFD induces loss of HSC expressing a high level of lipid rafts.** LR are small platforms that float freely within the lipid-bilayer of cell membranes and are composed of sphingolipids and cholesterol in the inner cytoplasmic leaflet of the lipid bilayer. A perturbation in the lipid composition of cells can modify the dynamic assembly of proteins and lipids in LR[19]. We therefore decided to study the direct impact of a short term HFD on hematopoiesis in comparison with a control diet (CD; 4% kJ of fat). Accordingly, we fed mice a modified diet with a 42% fat content for a period of 4 weeks. This HFD led to the loss of LR^hi cells only among HSC (LSK-CD34⁻) and early progenitors (LSK), while LR^hi cells among more mature progenitors (such as

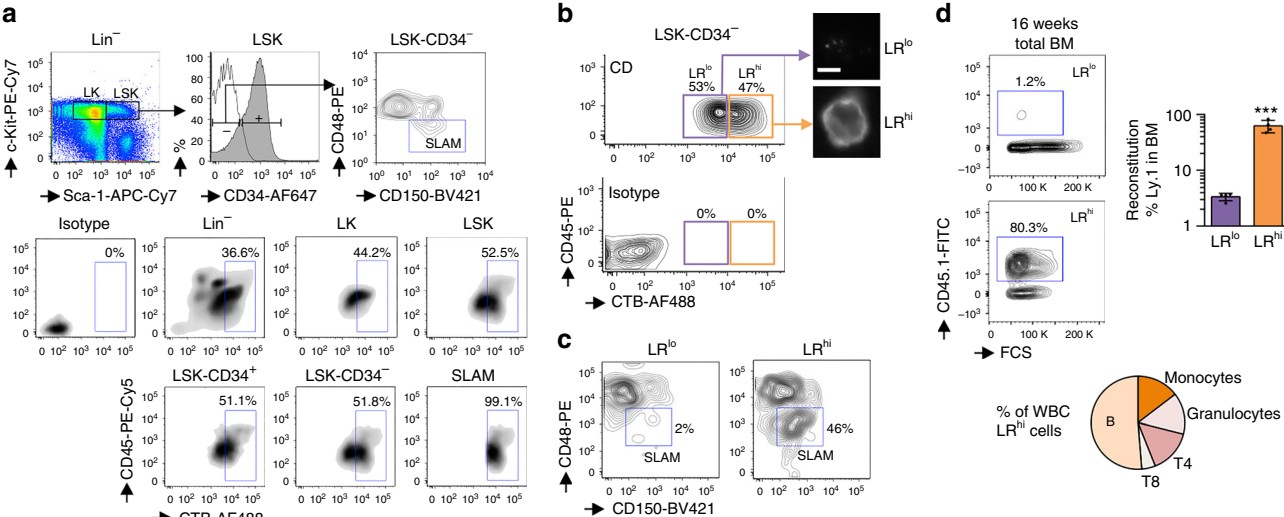

**Fig. 1** HSC displays high level of lipid rafts. **a** HSC from BM contains a high amount of lipid rafts (LR), following staining by flow cytometry with the cholera toxin subunit B (CTB). **b** Two distinct populations can be observed among LSK-CD34⁻ cells; one displaying high level (LR^hi) and the second one low level (LR^lo) of LR. Microscopy after cell sorting confirms the differential expression of LR between the two populations. White scale bar represents 5 μm. Data are representative of 4 mice. **c** LSK-CD34⁻ cells displaying high level of LR (LR^hi) are enriched in SLAM (CD150⁺ CD48⁻). Data are representative of 4 mice. **d** LR^hi cells are enriched in long-term reconstituted HSC, as assessed by relevant BM and WBC hematopoietic reconstitutions in secondary mice, 16 weeks after transplantation of 1000 LR^lo or LR^hi cells, *n* = 4 recipient mice. Data show mean ± SD; ns, non-significant (*P* > 0.05); ***P* < 0.001 (two-tailed unpaired Student's *t*-test)

Lin⁻ and Lin⁻ c-Kit⁺ (LK)) were not affected (Fig. 2), meaning that a HFD may disturb the maintenance of hematopoiesis in primitive HSC.

**Maintenance of primitive HSC is affected when mice are fed a HFD.** Food consumption was identical for CD and HFD over the 4-week period (~3 g/day/mouse, Supplementary Fig. 1a). While the HFD led to an increase in plasma LDL-cholesterol levels (Supplementary Fig. 1b), the short feeding period did not induce

major weight gain (Supplementary Fig. 1c). Furthermore, 6-h fasting blood glucose testing suggested an absence of prediabetes in the HFD mice (Supplementary Fig. 1d). When we investigated the impact of the HFD on hematopoiesis in BM, we detected a marked decrease in the number of HSC (LSK-CD34⁻) and the SLAM population (Fig. 3). Conversely, we found an increase in the LSK population, which was enriched in early progenitors (Fig. 3), but there was no impact on the distribution of other mature progenitors such as mega-erythroid progenitor (MEP), common myeloid progenitor (CMP), granulocyte/macrophage

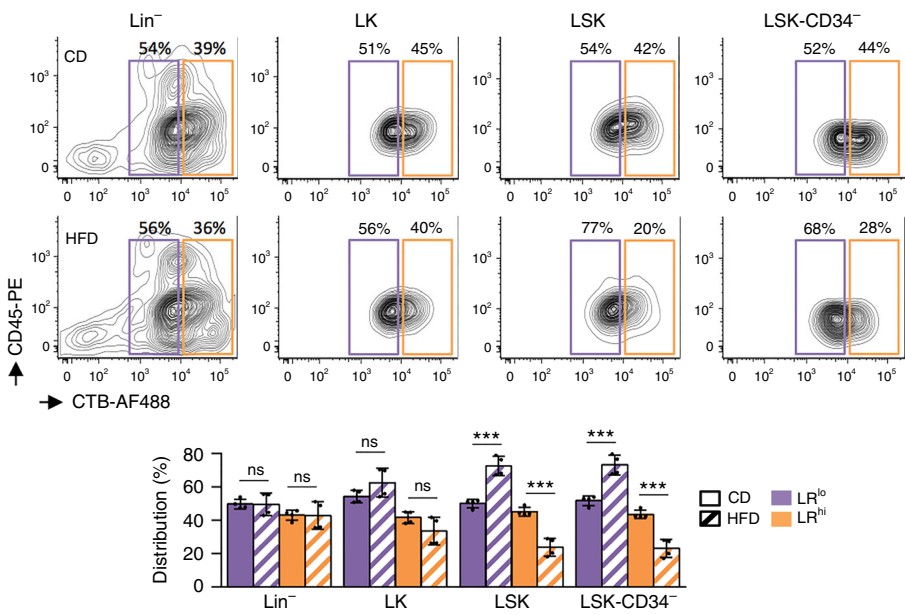

**Fig. 2** HFD affects the population of HSC expressing high level of lipid rafts in BM. The population of cells displaying a high level of lipid rafts (LR) (LR^hi) among LSK-CD34⁻ cells decreases when mice are fed with a HFD. Data show results on HSC (LSK-CD34⁻), early progenitors (LSK) and more mature progenitors (Lin⁻ and LK). Data show mean ± SD; n = 4 mice per diet group; ns, non-significant (P > 0.05); ***P < 0.001 (two-tailed unpaired Student's t-test)

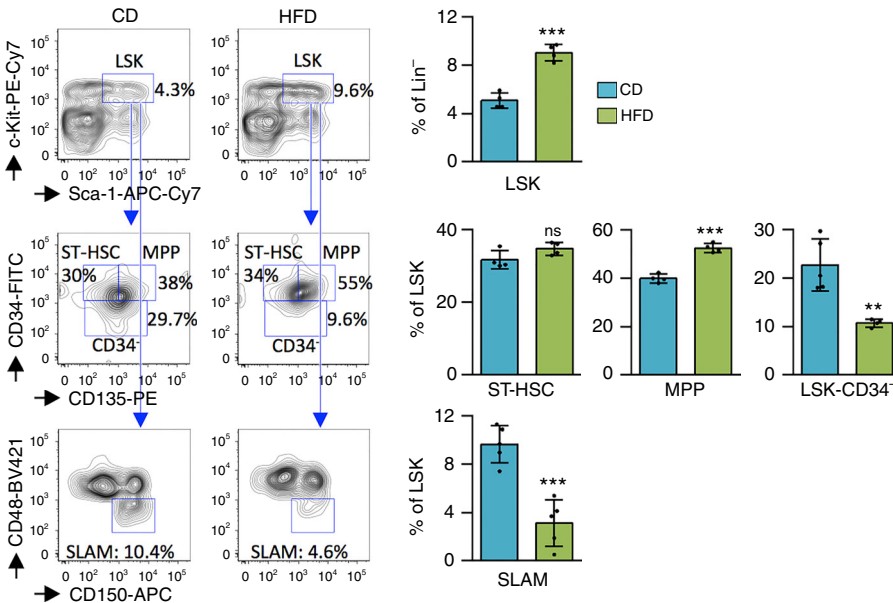

**Fig. 3** Short-term HFD impairs maintenance of HSC. Lin⁻ cells from 4-week HFD-fed mice are enriched in LSK cells and multipotent progenitors (MPP). HFD induced the loss of SLAM (CD150⁺ CD48⁻) HSC in BM. Data show mean ± SD (mice per diet group: n = 4 (CD) and n = 4 (HFD) for LSK, ST-HSC, and MPP; n = 5 (CD) and n = 4 (HFD) for LSK-CD34⁻; n = 5 (CD) and n = 5 (HFD) for SLAM); ns, non-significant (P > 0.05); **P < 0.01; ***P < 0.001 (two-tailed unpaired Student's t-test)

progenitor (GMP), and common lymphoid progenitor (CLP) (Supplementary Fig. 2a, b). In peripheral blood (PB), the HFD had no detectable hematological consequences on leukocyte, erythrocyte, and platelet count, or on hematocrit and hemoglobin levels (Supplementary Fig. 2c). Therefore, only primitive HSC were affected by the HFD.

**Loss of HSC following a HFD is dependent on intrinsic alteration.** To confirm that HFD affects maintenance of HSC, we performed competitive transplantation assays. Total BM isolated from CD-fed Ly.2 mice and HFD-fed Ly.1 mice (200,000 cells each) was transplanted into lethally irradiated mice (Fig. 4a). The HFD affected HSC homeostasis in primary mice, and the total BM cells isolated from HFD-fed mice exhibited lower long-term reconstitution potential in PB and BM, 16 weeks after the transplantation. Indeed, the BM of HFD-fed Ly.1 mice only reconstituted below 25% of the hematopoiesis system (Fig. 4b, c). Furthermore, we observed that BM cells isolated from HFD-fed mice reconstituted more mature PB cells in the myeloid lineages (monocytes and granulocytes), with a reduced reconstitution of T-lymphocytes, while the B-lymphocytes were unaffected. When we transplanted 1000 SLAM from CD-fed or HFD-fed Ly.1 mice in competition with 200,000 support BM Ly.2 cells, they exhibited equal long-term reconstitution potential in PB and BM, 16 weeks after the transplantation (Supplementary Fig. 3). Therefore, HFD only affected the number of HSC in primary recipient mice, but did not have an impact on the potential of the remaining HSC to reconstitute hematopoiesis.

In the obese, the accumulation of fat cells (adipocytes) within the BM has been shown to impair HSC functions, highlighting the role of the niche on homeostasis of HSC[15,23,24]. As assessed using paraffin-embedded, hematoxylin and eosin-stained BM, short-term intake of a HFD had no effect on the structure of the BM, with the same number of adipocytes observed and no change in size (Supplementary Fig. 4a–c). Nevertheless, in order to study how a HFD may affect the ability of the BM niche to retain HSC reconstitution potential, we fed Ly.2 host mice with a CD or HFD for 4 weeks and then lethally irradiated the animals prior to transplantation with fresh Ly.1 BM cells (200,000 cells). Both CD and HFD-fed mice exhibited the same ability to reconstitute hematopoiesis in PB and BM, 16 weeks after the transplantation

(Supplementary Fig. 4d–f). In conclusion, the HFD did not affect the ability of the BM niche to engraft HSC, and the loss of HSC can only be the result of loss of maintenance through an intrinsic mechanism.

**HFD alters the TGF-β mediated quiescence of HSC.** Considering that the majority of HSC are quiescent in homeostatic conditions[25,26], we next evaluated the effect of the HFD on cell cycles. Using the Ki67 proliferation marker and DAPI staining, we observed a decrease in the proportion of cells in G0 among LSK-CD34⁻ primitive HSC, indicating that the HFD had promoted the re-entry of HSC into the cell cycle (Fig. 5a). This was further confirmed by a CFU assay, which showed that HSC from HFD-fed mice produced colonies more rapidly than did control HSC on semisolid medium (Fig. 5b). Therefore, loss in maintenance of HSC following a HFD is linked to altered dormancy.

Previous studies have shown that TGF-β signaling is involved HSC subtype modulation[27] and quiescence[28–30], partly by preventing HSC re-entry into the cell cycle[21]. When we analyzed the transcription of different genes involved in HSC homeostasis, LSK-CD34⁻ from HFD-fed mice displayed reduced transcription for $p21^{Cip1}$ ($P < 0.01$; t-test) and $p57^{Kip2}$ ($P < 0.05$; t-test) *cyclin-dependent kinase inhibitor* genes known to be upregulated by TGF-β for quiescence of HSC[21,31], as well as an increased transcription of *c-Myc* ($P < 0.001$; t-test) inactivated by the TGF-β[32,33], while the transcription of several key transcription factors was not perturbed (Fig. 5c). Using flow cytometry, we furthermore observed a relevant decrease in the phosphorylation of Smad2/3 (pSmad2/3), downstream from the TGF-β receptor pathway[30] (Fig. 5d) in HSC from HFD-fed mice, while the phosphorylation of other important proteins (Akt, Stat3, or Stat5) were not affected by the exposure to a HFD (Supplementary Fig. 5). Hence, the TGF-β signaling-mediated quiescence of HSC seems to be specifically disturbed by a HFD.

**HFD affects quiescence of HSC due to LR/TGF-β disturbance.** We found that HSC expressing higher levels of LR were more affected following a HFD (Fig. 1). LRhi cells showed a more significant activation of the TGF-β pathway as assessed by a correlation between CTB and pSmad2/3 levels on LSK-CD34⁻ cells from CD mice, and, in addition, HFD induced a loss of LRhi/pSmad2/3⁺

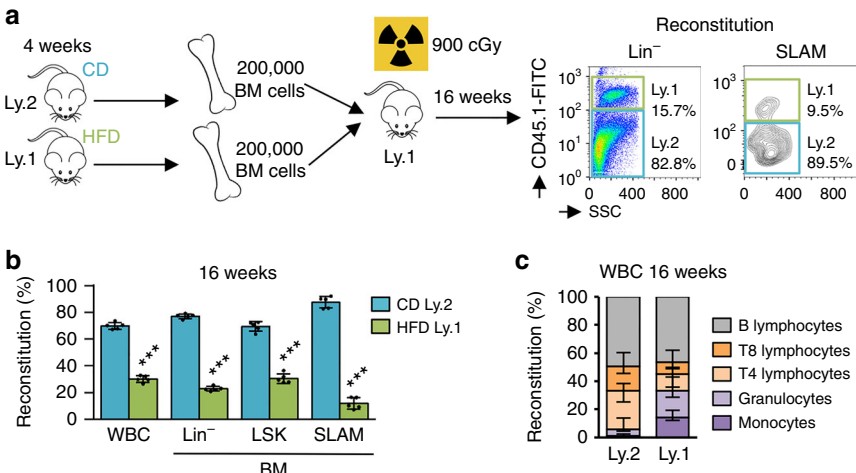

**Fig. 4** Erosion of HSC from BM after HFD follows a cell-autonomous response. **a** Experimental workflow. **b** Competitive transplantation in lethally irradiated recipient mice showing reduced reconstitution of HSC from Ly.1 HFD-fed mice, compared with HSC from Ly.2 CD-fed mice, 16-weeks after the transplantation. Reduced reconstitution is characterized in WBC and different subpopulations of the primitive hematopoietic compartment in BM, $n = 5$ recipient mice. **c** HSC from Ly.1 HFD-fed mice reconstitutes more in monocytes and granulocytes, with a reduced reconstitution of T4 and T8 lymphocytes, $n = 5$ recipient mice. Data show mean ± SD; ***$P < 0.001$ (two-tailed unpaired Student's t-test)

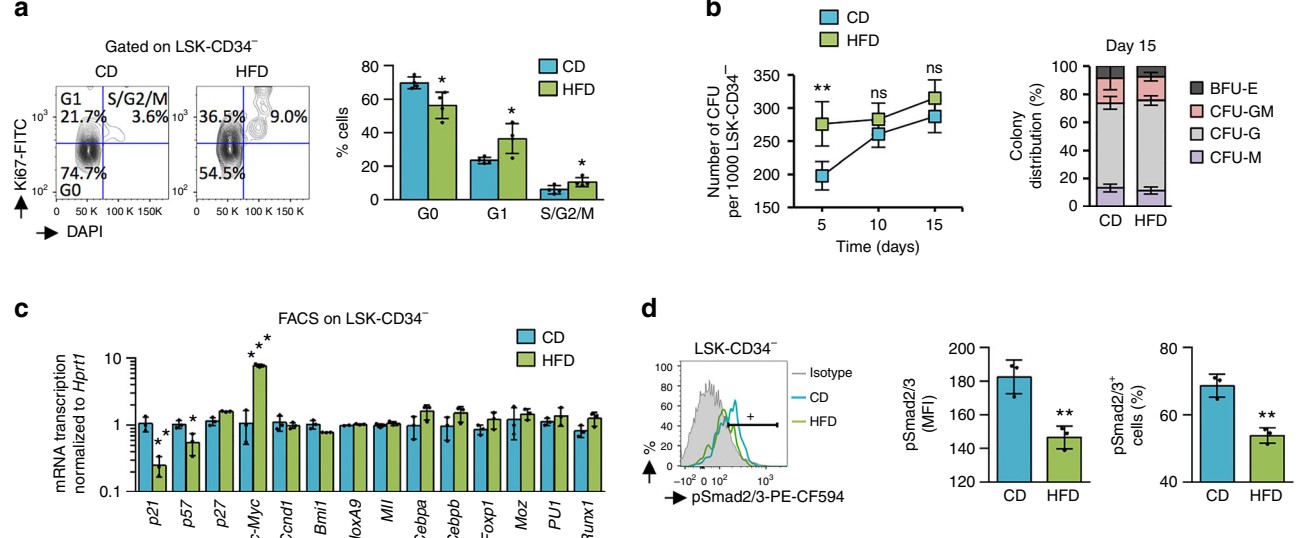

**Fig. 5** Short-term HFD affects the TGF-β mediated quiescence of HSC. **a** HFD affects quiescence of LSK-CD34− HSC, as assessed by the reduced proportion of cells in G0, $n = 4$ mice per diet group. **b** HFD HSC show quick cell forming unit (CFU)-colony formation on semisolid medium, which indicated that HFD HSC have left the dormant state and are going into differentiation more rapidly, $n = 4$ mice per diet group. As assessed on semisolid medium at day 15, no difference is observed among CFU; erythrocyte (E), granulocyte-monocyte (GM), granulocyte (G), and macrophage (M). **c** Target genes, known to be activated by TGF-β, such as the Cyclin-dependent kinase inhibitor $p21^{Cip1}$ and $p57^{Kip2}$ are downregulated in LSK-CD34− cells following a HFD. Furthermore, transcription of the *c-Myc* gene, repressed by TGF-β, is activated in HSC from HFD-fed mice, $n = 3$ mice per diet group. **d** HFD affects phosphorylation of pSmad2/3 in LSK-CD34− HSC. The pSmad2/3 is detected by flow cytometry and results show its expression (MFI; median fluorescence intensity) and proportion (%) of pSmad2/3+ cells among LSK-CD34− HSC, $n = 3$ mice per diet group. Data show mean ± SD; ns, non-significant ($P > 0.05$); *$P < 0.05$; **$P < 0.01$; ***$P < 0.001$ (two-tailed unpaired Student's $t$-test)

cells (Fig. 6a). We therefore decided to analyze the LR structure on the cell surface of HSC among LSK-CD34− LR^hi cells. When LR^hi cells were observed under the microscope following a HFD, LR organization was modified. The LR had more diffused distribution on HSC from CD-fed mice, whereas they appeared more clustered and formed larger platforms on HSC from HFD-fed mice (Fig. 6b), which can exhibit one, two, three, or more LR macrodomains (Supplementary Fig. 6). As a HFD might have perturbed the LR dynamic assemblies of proteins on HSC, we did a microscopy analysis of the TGF-β receptor 1 (Tgfbr1, ALK5) localization on the cell surface. Interestingly, the Tgfbr1 was more strongly condensed in LR clusters on LR^hi cells isolated from LSK-CD34− cells taken from HFD-fed mice ($P = 0.0426$; $t$-test) (Fig. 6c). While HFD did disturb clustering of Tgfbr1 within LR, flow cytometry revealed that HFD did not change expression of the Tgfbr1 receptor on the surface of LSK-CD34− cells (Supplementary Fig. 7). We also investigated whether HFD might affect the localization of several other membrane receptors (c-Kit, IL3Rα, and IL6Rα) involved in cytokine signaling in HSC, but found no perturbation when we analyzed LR organization and protein localization by microscopy (Supplementary Fig. 8).

**Injection of TGF-β1 in HFD-fed mice prevents loss of HSC.** In an attempt to rescue the phenotype, we injected 1.3 µg/kg of recombinant TGF-β1 (rTGF-β1) twice per week into the tail vein of mice while they were on the diet. The injection of rTGF-β1 led to the compensation of HFD-mediated HSC depletion ($P = 0.0057$; $t$-test) (Fig. 7a), which confirmed that loss of HSC when mice were fed a HFD is due to TGF-β signaling alteration. We also assessed whether rTGF-β1 injection compensated HSC function in transplantation assays (Fig. 7b). The findings supported our hypothesis that the disruption of lipid raft/TGF-β signaling leads to a loss of HSC function after a short-term HFD.

In addition, BM cells isolated from mice fed with a HFD and injected with rTGF-β1 can reconstitute hematopoiesis in BM (Fig. 7c) and PB (Fig. 7d) of secondary recipients, similar to the reconstitution in BM cells isolated from CD-fed mice. When mice were fed a HFD, the injection of rTGF-β1 did not change the distribution of LR clustering on the cell surface of LSK-CD34− cells. Meanwhile, we continued to observe the clustering of Tgfbr1 within LR, as assessed by immunofluorescence microscopy (Fig. 8). However, we discovered a relevant recovery in the phosphorylation level of Smad2/3 using immunofluorescence microscopy and flow cytometry (Fig. 9) after injection of rTGF-β1. Therefore, when HSC were isolated ex vivo and cell division was assessed with an in vitro system[28,34], HSC isolated from HFD-fed mice showed an increased proportion of cycling cells that, additionally, exhibited the ability to divide more rapidly (Supplementary Fig. 9a–b). In addition, the injection of rTGF-β1 in HFD-fed mice decreased HFD-mediated division in LSK-CD34− LR^hi HSC (Supplementary Fig. 9a–b), and rTGF-β1 treatment of freshly isolated HSC cells inhibits cell growth for each diet condition (Supplementary Fig. 9c), indicating that the HFD did not influence cell sensitivity to cytokine signals. Finally, a relevant recovery of the quiescence cell distribution (Supplementary Fig. 10) was also observed after the injection of rTGF-β1 in HFD-fed mice.

Therefore, we can conclude that injecting rTGF-β1 prevents a HFD-induced loss of HSC by reactivating the pSmad2/3-dependent TGF-β signaling involved in quiescence/maintenance of HSC, but without changing LR clustering on those particular HSC.

## Discussion

Several studies have described the essential role of lipids in hematopoiesis. A breakdown in HSC homeostasis contributes to inflammation in obese low-density lipoprotein receptor knockout mice[35]. Obesity activates myeloid cell production from BM

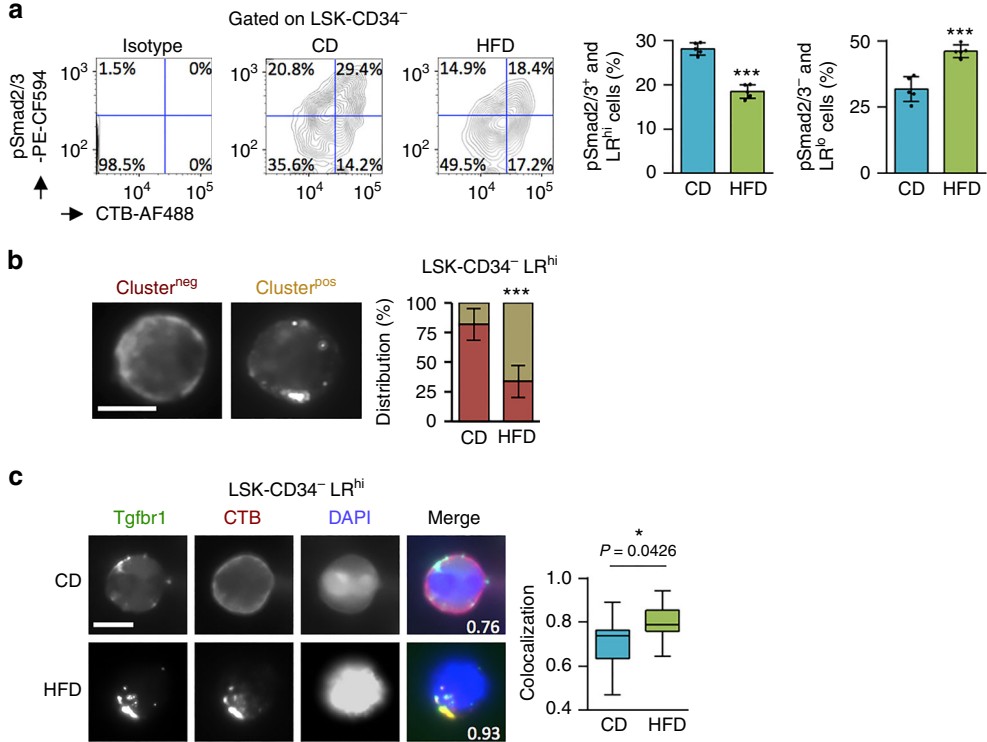

**Fig. 6** Short-term HFD disturbs lipid raft/TGF-β signaling on HSC. **a** LR[hi] HSC shows important phosphorylation of Smad2/3 (pSmad2/3) in CD-fed mice, and a HFD affects pSmad2/3 expression among HSC. The pSmad2/3 and LR (CTB) are detected by flow cytometry and results show the distribution (%) of pSmad2/3[+]/LR[hi] and pSmad2/3[−]/LR[lo] cells gated on LSK-CD34[−] HSC, n = 5 mice per diet group. **b** Single-cell immunostaining is performed on LSK-CD34[−] LR[hi] cells isolated by FACS. Data show clustering of LR (CTB) when mice are fed a HFD, n = 4 mice per diet group. **c** Single-cell immunostaining on LSK-CD34[−] LR[hi] cells showing that HFD induces cluster formation of LR (CTB, red), in which Tgfbr1 (green) is typically condensed. Data show examples of representative single-cell immunostaining for CD and HFD-fed mice, n = 4 mice per diet group. Statistics of the colocalization on LSK-CD34[−] cells (n > 30 cells) is shown on the right panel. Microscopy is performed on LSK-CD34[−] LR[hi] cells isolated by FACS. White scale bars represent 5 μm. **a**, **b** Data show mean ± SD. **c** Data are presented as median (central line), first and third quartiles (bottom and top of boxes, respectively), and whiskers (extreme values); *P < 0.05; ***P < 0.001 (two-tailed unpaired Student's t-test)

progenitors to potentiate inflammatory responses in metabolic tissues[17], and inflammation is known to affect the homeostasis of HSC, seeing as IFNα activates dormant HSC in vivo[36]. However, in our study, feeding mice a HFD over a period of 4 weeks had no consequences on inflammation; there was no modulation of pro-inflammatory (TNF-α, IL-1β, CCL2) or anti-inflammatory (TGF-β1, IL-10) cytokines detected in BM by *enzyme-linked immunosorbent assay* (Supplementary Fig. 11a), and inflammatory monocytes (Ly6G[−] Mac1[+] Ly6C[hi]) were not found to be increased in BM (Supplementary Fig. 11b).

Lipid intake can produce microenvironment-dependent defects which can perturb stem and progenitor cells[37]. For instance, HFD in pregnancy has been reported to compromise fetal hematopoiesis[16]. Lipids can also affect hematopoiesis through modulation/alteration of the support cells present in the BM microenvironment. For example, a mouse with deficiencies in the ATP binding cassette transporters ABCA1 and ABCG1 displayed a dramatic increase in HSC and progenitor cell mobilization and extramedullary hematopoiesis[38]. This is consecutive to elevated serum levels of G-CSF due to the generation of IL-23 by splenic macrophages and dendritic cells, which favors hematopoietic lineage decisions toward granulocytes rather than macrophages in the BM, leading to impaired support for osteoblasts and decreased specific cytokine production by mesenchymal progenitors. Obesity has also been described as affecting the homeostasis of HSC through a disturbance in the BM microenvironment. For example, obesity has been shown to suppress B lymphopoiesis by disrupting the supportive capacity of the IL-7 secretion mediated by supportive cells in the BM niche[15]. In obese mice fed a HFD, an increased number of adipocytes has been observed in BM which enhances hematopoiesis[18]. Various murine models of obesity have furthermore suitably shown that adipose tissue macrophages in obesity can promote proliferation of BM myeloid progenitors[24]. More recently, the interactions between adipocytes and HSC in BM have been identified[39], and the accumulation of adipocytes has been found to impair HSC function[23]. In the present study, we fed mice with a HFD only for a short period and the mice were not obese. To study if a HFD can influence the niche reconstitution within 4 weeks, BM cells isolated from CD-fed mice were transplanted into HFD-fed recipient mice. However there was no detectable defect in myeloid or lymphoid reconstitution. Conversely, transplantation of BM cells from HFD-fed mice in CD-fed recipient mice reproduced a defect in the reconstitution of hematopoiesis. These experiments prove that loss of HSC following a HFD, even after a short period of 4 weeks, is essentially dependent on intrinsic alteration rather than niche perturbation.

Nagareddy et al. have also described an expansion of hematopoietic stem and progenitor cells (HSPC; LSK cells) observed directly in a murine model of obesity, as well as in transplantation of obese BM into lean wild type mice recipients[24]. In another study Van den Berg et al. also described that diet-induced obesity in mice following 18-weeks induced a shift in HSC toward

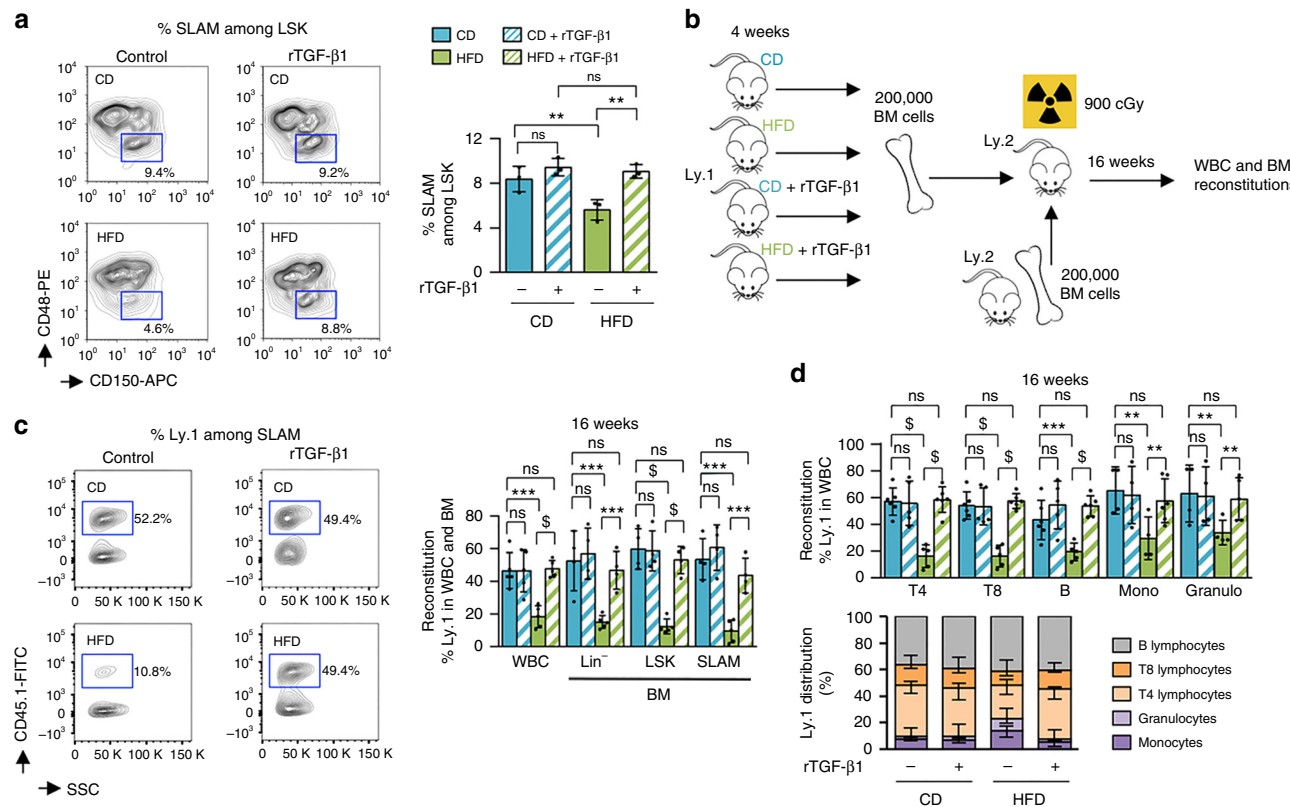

**Fig. 7** Injection of recombinant TGF-β1 in mice compensates for the loss of HSC following a HFD. **a** Recombinant TGF-β1 (rTGF-β1) is injected at 1.3 μg/kg twice per week into the tail vein of mice during the 4 weeks of the CD or HFD, and erosion of HSC is compensated, $n = 3$ mice per experimental group. **b** Experimental workflow to show that rTGF-β1 injection also compensates for HSC function in transplantation assays. **c** Injection of rTGF-β1 in HFD-fed mice improves BM's ability to reconstitute, 16-weeks after transplantation in lethally irradiated recipients (mice per experimental group: $n = 5$ (CD, CD + rTGF-β1, HFD, HFD + rTGF-β1) for WBC; $n = 4$ (CD, CD + rTGF-β1, HFD + rTGF-β1) and $n = 5$ (HFD) for Lin−, LSK, and SLAM. **d** Reconstitution is further improved in BM (mice per experimental group: $n = 6$ (CD, HFD + rTGF-β1) and $n = 5$ (CD + rTGF-β1, HFD) for T4, T8, and B; $n = 4$ (CD, CD + rTGF-β1, HFD) and $n = 6$ (HFD + rTGF-β1) for monocytes (Mono); $n = 4$ (CD, CD + rTGF-β1, HFD) and $n = 5$ (HFD + rTGF-β1) for granulocytes (Granulo). Data show mean ± SD; ns, non-significant ($P > 0.05$); *$P < 0.05$; **$P < 0.01$; ***$P < 0.001$; $, $P < 0.0001$ (two-tailed unpaired Student's *t*-test)

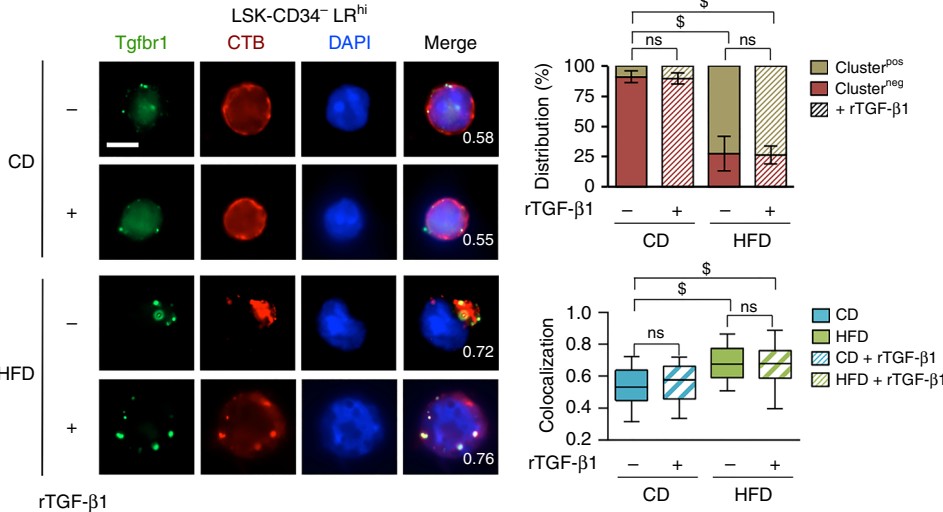

**Fig. 8** rTGF-β1 does not changed Tgfbr1 clustering within LR on LSK-CD34− LRhi cells. Single-cell immunostaining is performed and data show that HFD induces cluster formation of LR (CTB, red), in which Tgfbr1 (green) is typically condensed, and that rTGF-β1 injection do not change the redistribution of LR clustering. Data on the left panel show examples of representative single-cell immunostaining for CD and HFD-fed mice, treated or not by recombinant TGF-β1 (rTGF-β1 injected in vivo at 1.3 μg/kg) twice per week during the 4 weeks of the diet, $n = 2$ mice per experimental group. Microscopy is performed on LSK-CD34− LRhi cells isolated by FACS. White scale bar represents 5 μm. Graph and box-and-whisker plots, showing statistics of the distribution of LR clustering and colocalization on LSK-CD34− LRhi cells ($n > 40$ cells) respectively, are shown on the right panel. For top right graph, data show mean ± SD; for bottom right box-and-whisker plots, data are presented as median (central line), first and third quartiles (bottom and top of boxes, respectively), and whiskers (extreme values); ns, non-significant ($P > 0.05$); $, $P < 0.0001$ (two-tailed unpaired Student's *t*-test)

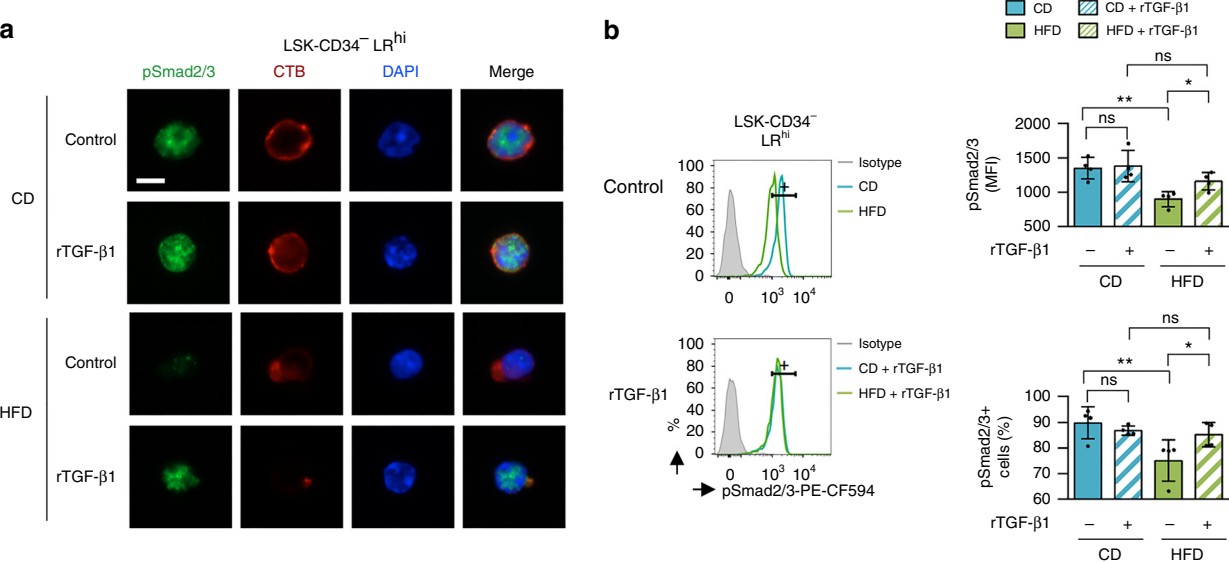

**Fig. 9** rTGF-β1 attenuates HFD-mediated inactivation of pSmad2/3 in LSK-CD34⁻ LR^hi HSC. **a** Single-cell immunostaining is performed on LSK-CD34⁻ LR^hi HSC isolated by FACS and using an anti-pSmad2/3 (green), CTB (red), and DAPI (blue). Data show examples of representative single-cell immunostaining for CD and HFD-fed mice, treated or not by recombinant TGF-β1 (rTGF-β1 injected in vivo at 1.3 µg/kg) twice per week during the 4 weeks of the diet, *n* = 2 mice per experimental group. White scale bar represents 5 µm. **b** The pSmad2/3 signal in LSK-CD34⁻ LR^hi HSC from CD and HFD-fed mice, treated or not by recombinant TGF-β1 (rTGF-β1 injected in vivo at 1.3 µg/kg) twice per week during the 4 weeks of the diet, is detected by flow cytometry. Results show their expression (MFI; median fluorescence intensity) and proportion (%) of pSmad2/3⁺ cells among LSK-CD34⁻ LR^hi HSC, *n* = 4 mice per experimental group. Data show mean ± SD; ns, non-significant (*P* > 0.05); **$P$ < 0.01 (two-tailed unpaired Student's *t*-test)

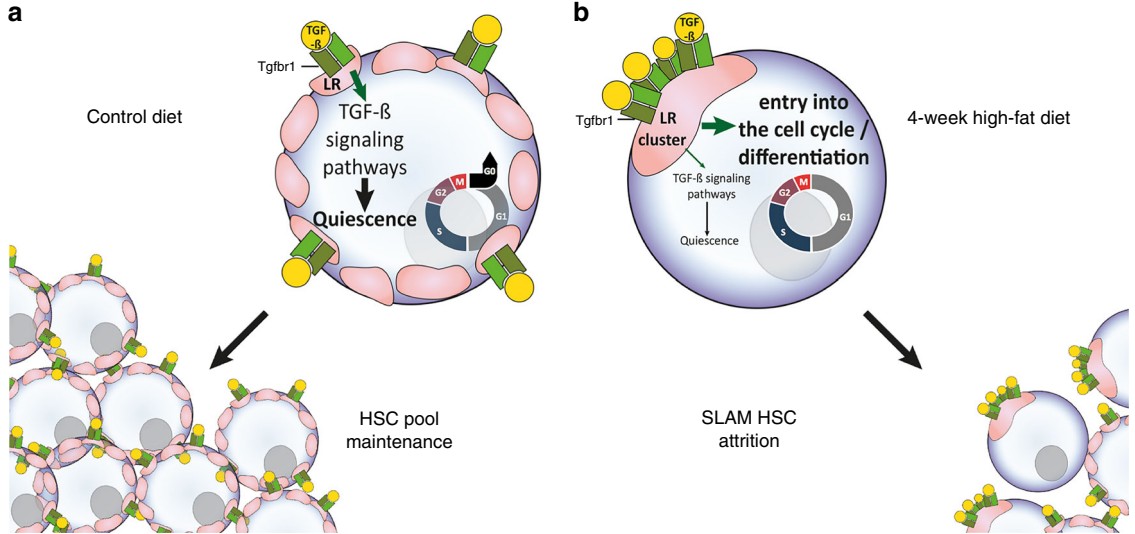

**Fig. 10** Schematic illustration of the proposed mechanism for erosion of HSC following a HFD. **a** Lipid raft (LR) staining is found spread on the cell surface of HSC isolated from control diet (CD)-fed mice. **b** In mice fed a high-fat diet (HFD), a clustering of LR is detected on HSC within 4 weeks. After a HFD, TGF-β receptor 1 (Tgfbr1) is more condensed within LR on HSC, which affects TGF-β stimulated quiescence of HSC

maturing multipotent progenitor (MPP) cells[14]. Our study corroborates the increased expansion of progenitors, but we highlight here that this is due to an exhaustion of the most primitive HSC (LSK-CD34⁻ or SLAM cells). Our study also shows that this switch from quiescent to differentiating cells might not be only related to an obesogenic environment, because, within as little as 4 weeks, there is an exhaustion of HSC in the BM of mice fed a HFD.

Several previous studies have shown that a HFD induces major perturbations in murine HSC and in homeostasis of the hematopoietic system. These alterations are frequently in relation to the pathophysiology of obesity after the extended consumption of a HFD over several weeks/months[15–18] or observed on mice

models with gene deficiencies to study obesity and the influence on hematopoiesis[24,35,38]. Our study revealed that a short-term HFD can rapidly (within 4-weeks) generate exhaustion of HSC in mouse BM, and we more fully described the direct impact that a HFD can have on LR organization on the cell surface of HSC in mouse BM. LR staining was found spread across the surface of HSC isolated from CD-fed mice, while LR were clustered on HSC isolated from mice fed a HFD. The TGF-β receptor 1 (Tgfbr1) was found more condensed within LR on HSC after a HFD, which affects TGF-β stimulated quiescence of HSC (Fig. 10). Our data confirms a previously described study indicating that LR clustering is essential for HSC emergence from quiescence and re-entry into the cell cycle[21]. While Yamazaki et al. have reported

that LR clustering induced by cytokine stimulation is critical for promoting HSC and progenitor cell division, our study confirms that LR reorganization is indispensable for HSC to emerge from hibernation and to re-enter the cell cycle, and that HFD induces LR clustering in HSC and decreases downstream TGF-β signaling pathways, as reflected by reduced phosphorylation of the Smad2/3 downstream pathway, an important determinant in TGF-β signaling[30], and reduced or increased transcription of genes known to be upregulated ($p21^{Cip1}$ and $p57^{Kip2}$) or inactivated (c-Myc) by TGF-β, respectively. Our findings highlight that HFD-induced recruitment and concentration of Tgfbr1 signal transducers into LR clusters yield inefficient and impaired transduction of TGF-β-mediated signals leading to HSC quiescence. We further assessed other key receptors known to be regulated by LR[19–22], such as c-Kit, IL3Rα, and IL6Rα, but failed to observe changes in their LR clustering on HSC following HFD (Supplementary Fig. 8).

Moreover, a HFD led to the loss of LR^hi cells among HSC (LSK-CD34−), whereas LR^hi cells among more mature progenitors (such as LK and Lin−) were not affected. Interestingly, LSK-CD34− cells expressed more Tgfbr1 (Supplementary Fig. 7), so, consequently, we can suggest that the LR/TGF-β signaling is probably more sensitive to HFD intake. Finally, injection of recombinant TGF-β1 in HFD-fed mice twice weekly for 4 weeks prevents HSC loss and the alteration of BM's ability to reconstitute. This injection of recombinant TGF-β1 into mice under a HFD regimen reactivates the Smad2/3 downstream pathway in HSC, indicating that HFD-induced HSC erosion is supported by alteration of TGF-β signaling.

Studies on the effects of a HFD on the homeostasis of human HSC and progenitors is rather difficult to conduct, and rodent models are therefore often used. The Western diet contains 42% fat and is therefore comparable to "fast food" which contains an estimated 35–45% fat. Our study has not only generated valuable insight into how stem cells are controlled by diet-dependent pathways, but also establishes that consuming a HFD for as little as 4 weeks can disturb the LR constituents on primitive HSC, leading to the alteration of the TGF-β signaling-mediated HSC quiescence and to exhaustion of HSC in mouse BM.

## Methods

**Mice**. C57BL/6J (Ly.2) and congenic B6.SJL (Ly.1) mice were provided by Envigo and kept in the Animal Facility at the University of Burgundy. Doses of 1.3 μg/kg of rTGF-β1 (7666-MB, R&D Biosystems) were injected twice per week into the tail vein (i.v.) of mice during the 4 weeks when mice were fed a HFD or a CD. The Ethics Committee for Animal Welfare of the University of Burgundy and the French Ministry of Higher Education and Research approved all animal experiments (references 01333.02 and 01318.02).

**Peripheral blood analyses**. Peripheral blood (PB) was collected from the tail vein, and erythrocytes were lysed with a hemolytic buffer (150 nM NH₄Cl, 10 mM KHCO₃, 0.1 mM ethylene-diamine-tetra-acetic acid). The remaining cells were stained with conjugated antibodies for flow cytometry analysis. After tail vein PB sampling, hematopoietic cells were counted using a hemocytometer (SCIL Vet ABC+, Oostelbeers, The Netherlands). After tail vein PB sampling and centrifugation (10,000×g, 10 min), plasma low density lipoprotein (LDL) was determined (Indiko, Thermo Fisher Scientific). Four weeks after the start of the diet, plasma glucose levels were measured in tail blood (Glucometer-One Touch Ultra) in mice fasted for 6 h.

**Food**. Control mice were maintained on an ad libitum 4% fat rodent chow diet (CD). 8 to 14-week-old mice were fed for 4 weeks with a 42% fat HFD (MD. 88137, Envigo RMS Division, Indianapolis, IN). The crude lipid extract from CD and HFD pellets was analyzed with mass spectrometry and results are shown in Supplementary Table 1. The procedure was performed by a Lipidomic Analytical Platform (Université Bourgogne-Franche-Comté, Dijon, France).

**BM analyses and transplantation**. Hind limb bones were crushed in a mortar and total BM cells were filtrated (30 μm). Magnetically lineage-depleted BM cells (Lineage Cell Detection Cocktail-Biotin, 130-092-613, Miltenyi Biotec) were

stained in phosphate-buffered saline (PBS), pH7.2, with combinations of antibodies for flow cytometry. For the transplantation study, Ly.2 and Ly.1 mice were fed with a CD or HFD, respectively, for 4 weeks and sacrificed for BM transplantation (200,000 cells). Cells were transplanted in competition into the tail vein of lethally irradiated (900 cGy) recipients. PB and BM reconstitutions were analyzed 16 weeks after the transplantation. Further information regarding transplantation were described on the supplementary Methods section.

**Flow cytometry and fluorescent-activated cell sorting (FACS)**. For the lineage staining in PB, CD4-PE-CF594 (562285, dilution ratio 1:100), CD8-AF647 (557682, dilution ratio 1:100), Mac1-APC-Cy7 (557657, dilution ratio 1:100), Mac1-PE-Cy7 (552850, dilution ratio 1:100), Mac1-AF647 (557686, dilution ratio 1:100), GR-1-FITC (553127, dilution ratio 1:100), GR-1-PE (553128, dilution ratio 1:100) antibodies (BD Biosciences) and CD19-PE (557399, dilution ratio 1:100), CD3-PB (558214, dilution ratio 1:100), B220-AF647 (103226, dilution ratio 1:100) antibodies (Biolegend) were used. Total BM cells and magnetically lineage-depleted BM cells (Miltenyi Biotec) were stained in PBS with combinations of the following antibodies conjugated to fluorochromes: c-Kit-PE-Cy7 (558163, dilution ratio 1:100), CD34-AF647 (560230, dilution ratio 1:50), CD34-FITC (553733, dilution ratio 1:50), CD135-PE (553842, dilution ratio 1:100), CD48-PE (557485, dilution ratio 1:50), CD48-BV421 (562745, dilution ratio 1:50), CD45-PE-Cy5 (561870, dilution ratio 1:100), CD45-PE (553081, dilution ratio 1:100), CD16/32-FITC (553144, dilution ratio 1:100), IL7Rα-PE-CF594 (562419, dilution ratio 1:100), Mac1-AF647 (557686, dilution ratio 1:100), Ly6C-PE-CF594 (562728, dilution ratio 1:100) (BD Biosciences) and c-Kit-PB (105820, dilution ratio 1:100), CD150-APC (115910, dilution ratio 1:100), CD150-BV421 (115925, dilution ratio 1:100), Sca-1-APC-Cy7 (108126, dilution ratio 1:100), Ly6G-FITC (127605, dilution ratio 1:100) (Biolegend). To separate donor cells from support and recipient cells, CD45.1 (Ly.1)-FITC (553775, BD Biosciences, dilution ratio 1:100) antibody was used. AF488- (C-34775, 1 μg/mL) or AF555- (C-34776, 1 μg/mL) conjugated cholera toxin subunit B (Thermo Fisher Scientific) were used to stain LR. For studies on cell cycle and quiescence, Ki67-FITC antibody (556026, BD Biosciences, dilution ratio 1:6) was used. Tgfbr1 was stained with anti-Tgfbr1-PE antibody (FAB5871P, R&D Systems, dilution ratio 1:100). For intracellular protein staining, anti-phospho-Smad2(S465/S467)/Smad3(S423/S425)-PE-CF594 (562697, BD Biosciences, dilution ratio 1:20), anti-phospho-Stat5(Y694)-PE-Cy7 (560117, BD Biosciences, dilution ratio 1:6), anti-phospho-Akt(S473)-APC (130-105-293, Miltenyi Biotec, dilution ratio 1:11) and anti-phospho-Stat3(Y705)-FITC (651019, Biolegend, dilution ratio 1:20) antibodies were used after cell surface staining and fixation/permeabilization using BD Cytofix/Cytoperm Plus Fixation/Permeabilization Kit (BD Biosciences). All events were acquired by a FACS Canto10 flow cytometer (BD Biosciences) equipped with BD FACSDiva software (BD Biosciences). LSK-CD34− cells were sorted on a FACS Aria cell sorter (BD Biosciences) equipped with BD FACSDiva software (BD Biosciences). Data were analyzed using FlowJo software (TreeStar Inc). Flow cytometry-gating strategies are presented in the Supplementary Figs. 12–17.

**Colony-forming unit (CFU) assays**. BM was isolated and $1 \times 10^4$ FACS-sorted LSK-CD34− cells were cultured in 3 mL semisolid methylcellulose medium supplemented with growth factors (Methocult M3434; Stem Cell Technologies) at 37 °C in 98% humidity and 5% CO₂ for 15 days. CFU were counted at different time points.

**Immunofluorescence and microscopy**. LSK-CD34− cells were stained with cell surface antibodies prior to cell sorting on glass slides and microscopy. For colocalization of LR and Tgfbr1, we used AF555-conjugated CTB (C-34776, Thermo Fisher Scientific, 1 μg/mL) and anti-Tgfbr1 (PA5-38718, Thermo Fisher Scientific, dilution ratio 1:100) with secondary anti-rabbit AF488 (A27034, Thermo Fisher Scientific, dilution ratio 1:1000) antibodies. For the phospho-Smad2/3 microscopy, cells were stained with the cell surface markers (LSK-CD34−) and AF555-conjugated CTB (C-34776, Thermo Fisher Scientific, 1 μg/mL), prior to fixation/permeabilization using BD Cytofix/Cytoperm Plus Fixation/Permeabilization Kit (BD Biosciences), then we stained with an anti-phospho-Smad2(S465/S467) antibody (AB3849, Merck, dilution ratio 1:250) and with secondary anti-rabbit AF488 (A27034, Thermo Fisher Scientific, dilution ratio 1:1000) antibody. Cells were sorted on glass slides and fixed with ProLong Gold Antifade reagent containing DAPI (P36931, Thermo Fisher Scientific). Images were acquired with an Axio Imager M2 (Zeiss) coupled with an Apotome.2 (×63 objectives) and processed for colocalization studies (Fiji, NIH software). Further information regarding immunofluorescence assays are described in the supplementary Methods section.

**Quantitative real-time (RT)-PCR**. After mRNA isolation from LSK-CD34− cells with the RNeasy kit (Qiagen), M-MLV reverse transcriptase (Promega) was used to synthesize cDNA according to the manufacturers' recommendations. The following TaqMan assays (Qiagen) were then used for qPCR: Mm00432448 (p21), Mm00438170 (p57), Mm00438168 (p27), Mm00487804 (c-Myc), Mm00432359 (Ccnd1), Mm03053308 (Bmi1), Mm00439364 (Hoxa9), Mm01179235 (Mll), Mm00514283 (Cebpa), Mm00843434 (Cebpb), Mm00474845 (Foxp1),

Mm01211940 (Moz), Mm004881α (PU1), Mm00486762 (Runx1), Mm03024075 (Hprt1). Samples were run in triplicate using the ABI ViiA 7 Real Time PCR System (Applied Biosystems, Foster City, CA).

**Statistics**. All measurements were taken from distinct samples. Animals were randomized to diet experimental groups. Data are expressed as a mean ± SD or represented in box-and-whisker plot format. The differences between experimental groups were assessed with two-tailed unpaired Student's *t*-tests (*t*-test). Statistics were done with Prism 6 software (GraphPad). A *P*-value of less than 0.05 was considered statistically significant and significance is indicated on the figures with the following symbols: $*P < 0.05$; $**P < 0.01$; $***P < 0.001$; $\$P < 0.0001$.

**Reporting summary**. Further information on experimental design is available in the Nature Research Reporting Summary linked to this article.

## Data availability
The data that support the findings of this study are available from the corresponding author upon reasonable request.

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

## Acknowledgements
This work was supported by a French Government grant managed by the French National Research Agency under the program 'Investissements d'Avenir' with reference ANR-11-LABX-0021 (LipSTIC LabEx), by the Conseil Régional de Bourgogne through the plan d'action régional pour l'innovation (PARI) and the European Union through the PO FEDER-FSE Bourgogne 2014/2020 programs, and the Association pour la Recherche Clinique et la Formation en Hématologie. F.H. was supported by a fellowship from the LipSTIC LabEx and from the European Union through the PO FEDER-FSE Bourgogne 2014/2020 programs. A.B. was supported by a fellowship from the French Ministère de la Recherche et de l'Enseignement Supérieur (MRES). The authors express their gratitude to the animal housing facility, lipidomic analytical platform, and cytometry platform staff at the University of Burgundy (Dijon, France), especially Valérie Saint-Giorgio, Hélène Choubley, Anabelle Sequeira, Arlette Hammann, Serge Monier and Tiffany Thevenard, for their excellent suggestions and valuable technical support. The authors are grateful to Lorène Lebrun and Guillaume Maquart (INSERM UMR1231) for their technical help and advice, to the CellImaP platform for histological experiments, as well as to Pr. Laurent Martin from the department of anatomo-pathology of the CHU of Dijon for numeration of slides, and to Suzanne Rankin (CHU Dijon) for editing and critical reading of this manuscript.

## Author contributions
F.H. performed experiments, analyzed data and wrote the manuscript; A.B. and A.A. helped with experiments; J.P.P.B. performed lipidomic analyses; J.N.B. and L.D. discussed the data; R.Q. conceived the study, performed experiments, analyzed data and wrote the manuscript.

## Additional information

**Competing interests:** The authors declare no competing interests.

