## [Peer Review File · Nature Communications]

Reviewers' comments:

Reviewer #1 (Remarks to the Author):

General comments:

The authors describe a short-term HFD disturb the lipid raft constituents on primitive HSC, leading to the alteration of TGF- β signaling-mediated HSC quiescence and to exhaustion of HSC in mouse BM. We are aware of the need for additional data for the current paper is well write and presented. However, some more add data is needed as well.

1) The authors mentioned that the injection of rTGF- β 1 into mice under a HFD regimen prevents HSC loss and BM reconstitution ability alteration. Authers should confirm the redistribution of lipid raft clustering after injection of rTGF- β 1 by immunofluorescence microscopy.

2) Furthermore, the authors should consider clarifying the downstream effects of TGF- β 1 signaling (i.e. lipid raft-PI3K-Akt-FOXO pathway). Individual points:

3) It is important to show the difference in cytokine signal between LR-lo and LR-hi. It should also be analyzed after HFD diet.

4) Is cell division of HSC after HFD normal? Please check with in vitro system. (Using TGF-beta+ or - conditions)

5) The bone marrow environment after HFD is expected to be highly fat cells. Authors need to consider whether niche influences.

Reviewer #2 (Remarks to the Author):

This is an interesting paper, and the experiments are well performed. This paper describes a first potential mechanism (TGFbeta) why a high fat diet perturbs HSC biology. However, although interesting, the novelty of the first part of the paper is limited, and also the reconstitution experiments, as well as the quiescence data are confirmative of data that have been published by others (i.e. van den Berg et al, Nagareddy et al).

- What part of TGFbeta signaling does not function?? On the level of ALK, or on the level of SMAD2/3 or SMAD7 signaling? Or does TGFbeta signaling divert to other pathways?

- The discussion does not take into account earlier published work, and does not highlight the similarities/differences with the current data

- Why does the high fat diet (42%) induce loss of primitive HSCs, whereas the high cholesterol diets increase the HSC population??

- Nagareddy et al (Cell metabolism 2014) shows different results than described in this paper, please discuss and explain the differences).

- what other lipid raft dependent pathways are disturbed? Only TGFbeta?

- What are the results of the lipid raft disturbances, loss of caveolin dependent endocytosis?

Reviewer #3 (Remarks to the Author):

Hermelet et al have asked whether high-fat diet influences the function of hematopoietic stem cells (HSC). The investigators studied lipid rafts in HSC and found that the most primitive HSC contain high levels of lipid rafts and that these HSC have better reconstitution capacity after bone marrow transplantation than HSC with lower levels of lipid rafts. The authors gave the mice high fat diet and found that this diet after only 4 weeks induced a loss in HSC with high levels of lipid rafts. Furthermore, the high fat diet induced reduced HSC function as measured by competitive transplantation. HSC from treated mice exhibit a lower fraction of quiescent HSC and target genes of TGF-beta showed reduce transcription (p21, p57, etc.). The distribution of lipid rafts in HSC was altered on high-fat treated HSC and the distribution of TGF-beta receptors within lipid rafts was altered as well. When high-fat treated mice were given TGF-beta1, loss of HSC and reduced reconstitution ability was prevented, suggesting that the high-fat diet induces alterations in lipid rafts that reduce or prevent the action of TGF-beta within HSC, thereby inducing a loss in quiescent HSC.

The experiments in this paper are well performed, the approach is thorough and the conclusions are sound. This represents original and innovative research where the authors also provide a mechanism for the findings. I have only minor criticism and suggestions as follows:

- 1) Page 4, paragraph about TGF-beta. ...homeostasis of HSC from HFD treated mice LSK-CD34-... The authors should clarify that these HSC are from HFD treated mice
- 2) Page 4 next paragraph.as LR had more diffused distribution on CD-HSC... The word distribution is missing
- 3) Same paragraph third line from bottom "change" instead of "changed"
- 4) Page 4, last paragraph. "In a way to recue the phenotype" must be meant to say "In an attempt to rescue the phenotype".
- 5) In the discussion it would be interesting to discuss why the authors think that the effect is so specific for HSC. TGF-beta acts powerfully on hematopoietic progenitors, not the least erythroid progenitors. Why is there no effect on the progenitors?
- 6) It would be interesting if the authors comment in the discussion on the dose of the high-fat diet. Is this dose much higher than humans will normally eat if they eat a lot of fat or is it similar? Are high-fat eaters likely to get reduced HSC function?

Reviewer #1's comments:

The authors describe a short-term HFD disturb the lipid raft constituents on primitive HSC, leading to the alteration of TGF- β 1 signaling-mediated HSC quiescence and to exhaustion of HSC in mouse BM. We are aware of the need for additional data for the current paper is well write and presented. However, some more add data is needed as well.

We would like to acknowledge the reviewer for the constructive review on our manuscript. To help track changes in the manuscript, all modifications have been marked in red in the main text as well as in the supplementary data and in figure legends.

- 1) The authors mentioned that the injection of rTGF- β 1 into mice under a HFD regimen prevents HSC loss and BM reconstitution ability alteration. Authors should confirm the redistribution of lipid raft clustering after injection of rTGF- β 1 by immunofluorescence microscopy.

To address the first comment from Reviewer 1, we acquired new sets of images by immunofluorescence microscopy showing lipid raft domains and Tgfr1 staining in LSK-CD34⁻ cells from CD and HFD-fed mice, treated or not by recombinant TGF- β 1. These images have been added in the new version of the manuscript (new Supplementary Fig.S9). Although no inhibition of LR clustering was observed by immunofluorescence microscopy, we did demonstrate a relevant rescue in the phosphorylation level of Smad2/3 by flow cytometry and immunofluorescence microscopy (new Supplementary Fig.S10), as well as quiescence cell distribution after injection of rTGF- β 1 (new Supplementary Fig.S12). Therefore, we can conclude that injecting rTGF- β 1 prevents a HFD-induced loss of HSC by reactivating the TGF- β signaling involved in quiescence/maintenance of HSC, but without changing LR clustering on these HSC.

- 2) Furthermore, the authors should consider clarifying the downstream effects of TGF- β 1 signaling (i.e. lipid raft-PI3K-Akt-FOXO pathway).

We performed new experiments showing that HFD affects phosphorylation of Smad2/3 (new Supplementary Fig.S10), a canonical phospho protein known to be important in TGF- β signaling. We performed another experiment (in the new Supplementary Fig.S5) showing that other important signaling pathways such as phospho-Akt, phospho-Stat3 or phospho-Stat5 were not perturbed after HFD exposure. We also assessed the localization of several other receptors (such as c-Kit, IL3R α and IL6R α) among lipid rafts and, following a HFD, we did not observe significant changes on the clusterization of these receptors (new Supplementary Fig.S8). These experiments suggest that HFD only disturbs Tgfr1 localization and the downstream phosphorylation of Smad2/3, a canonical effector of TGF- β signaling.

- 3) It is important to show the difference in cytokine signal between LR-lo and LR-hi. It should also be analyzed after HFD diet.

We added a new figure to the revised manuscript (see the new Fig.6a) to address the difference in cytokine signaling between LR^{lo} and LR^{hi}. Among LSK CD34⁻ HSC, LR^{hi} cells showed higher activation of the TGF- β pathway, as assessed by a relevant correlation between CTB and pSmad2/3 levels observed on LSK-CD34⁻ cells from CD mice. As asked by the reviewer, this signalling has been further analysed after HFD and we clearly observed disappearance of the LR^{hi}/pSmad2/3⁺ cells.

- 4) Is cell division of HSC after HFD normal? Please check with in vitro system. (Using TGF-beta+ or - conditions)

For this interesting point, we followed previously described protocols (Ema, H et al., 2006, Nature protocols; Yamazaki, S et al., 2009, Blood), and we isolated single cells by FACS ex vivo after HFD. The experiments (see the new Supplementary Fig.S11) confirm that HSC purified from HFD-fed mice went more rapidly into division. This experiment therefore confirms our idea that a HFD can affect the quiescence of HSC in vivo. This experiment was also conducted on *ex vivo* cells when mice were fed a HFD and recombinant TGF- β 1 were administered, and confirms that reactivation of the TGF- β pathway can maintain quiescence (low cell division) of HSC in vivo when mice were fed a HFD (new Supplementary Fig.S11a-b). Furthermore, the addition of recombinant TGF- β in vitro on HSC from CD and HFD-fed mice show that both conditions were sensitive to recombinant TGF- β activation of quiescence (low cell division) (new Supplementary Fig.S11c), which suggest that the TGF- β pathway can be reactivated on HSC purified from HFD-fed mice.

5) The bone marrow environment after HFD is expected to be highly fat cells. Authors need to consider whether niche influences.

Indeed, lipid intake can produce microenvironment-dependent defects (Wilkinson, AC & Yamazaki, S, 2018, International journal of hematology), and in different murine models (either fed with a HFD during a long period to induce obesity or on KO mice models used to study obesity), it has been shown that adipose tissue in obesity can promote the proliferation of bone marrow myeloid progenitors (Nagareddy, PR et al., 2014, Cell metabolism). More recently, the interactions between adipocytes and HSC in bone marrow have been identified (Zhou, B et al., 2017, Nature cell biology) and the accumulation of adipocyte has been found to impair HSC function in cases of obesity (Ambrosi, TH et al., 2017, Cell Stem Cell). However in our study, we are not working with obese mice and we fed mice with a HFD only for a short period. We added the information/references in the discussion section to better contextualize the current work within the literature and to provide a more complete version of the revised paper. Moreover, we performed another experiment to address the important concerns raised by reviewer 1 concerning the highly fat cell distribution in the bone marrow environment after a 4-week HFD. Using paraffin-embedded, hematoxylin and eosin-stained bone marrow, we observed that short-term feeding with HFD seems to have no effect on the structure of BM, with the same number of fat cells (adipocytes) observed by immunohistochemistry analysis, and no perturbation in their size (see the new Supplementary Fig.S4a-c). Therefore we explain that in a context of short-term feeding, the niche does not influence the phenotype. This is further confirmed by our transplantation experiments. Indeed, as shown in Supplementary Fig.S4d-f of the manuscript, the experiments performed to monitor the potential impact of niche on hematopoietic reconstitution after total bone marrow cells transplantation demonstrate that HFD exposure prior to transplantation does not impact hematopoietic reconstitution. This experiment suggests an absence of niche influences on HSC homeostasis and distribution under HFD condition.

Reviewer #2's comments:

This is an interesting paper, and the experiments are well performed. This paper describes a first potential mechanism (TGFbeta) why a high fat diet perturbs HSC biology. However, although interesting, the novelty of the first part of the paper is limited, and also the reconstitution experiments, as well as the quiescence data are confirmative of data that have been published by others (i.e. van den Berg et al, Nagareddy et al).

We would like to thank the reviewer for the constructive review of our manuscript. Please find below new experiments and discussion to exhibit how our study provides novel and original data. To help track changes, all modifications have been marked in red in the main text as well as in the supplementary data and in figure legends.

- 1) What part of TGFbeta signaling does not function? On the level of ALK, or on the level of SMAD2/3 or SMAD7 signaling? Or does TGFbeta signaling divert to other pathways?

We performed new flow cytometry and microscopy-based experiments (see new Fig. 5d and new Supplementary Fig.S5, and Fig.S10 of the new manuscript) showing that HFD affects phosphorylation of Smad2/3, the downstream phospho protein activated by the Tgfr1 receptor and known to be important in TGF- β signaling. Interestingly, we also demonstrated a relevant rescue in the phosphorylation level of Smad2/3 by flow cytometry and microscopy (new Supplementary Fig.S10) when rTGF- β 1 is injected during a HFD feeding period (new Supplementary Fig.S10). We performed another experiment (in the new Supplementary Fig.S5) showing that other important signaling pathways such as phospho-Akt, phospho-Stat3 or phospho-Stat5 were not perturbed after HFD exposure. So, our experiments suggest that a HFD only perturbs the phosphorylation of Smad2/3, a canonical effector of TGF- β signaling.

- 2) The discussion does not take into account earlier published work, and does not highlight the similarities/differences with the current data.

We reorganized the first part of the discussion, which now describes more similarities and differences with the current data. One paragraph concerns the inflammation and obesity and the fact that short-term feeding with HFD does not modify inflammation in mouse BM. The second paragraph describes the literature regarding the role of obesity and lipid intake with changes in the niche at different levels. The literature on this subject is rich, and we have included many interesting recent works (Wilkinson, AC & Yamazaki, S, 2018, International journal of hematology; Nagareddy, PR et al., 2014, Cell metabolism; Zhou, B et al., 2017, Nature cell biology) in the discussion section to better contextualize our work within the existing literature and to provide you with a more complete version of the revised paper. We also discuss our observations which revealed that short-term feeding does not induce obesity in mice, and our transplantation experiment which does not support the idea that 4 weeks of lipid intake can modify the micro environmental niche. Our transplantation experiment proves that loss of HSC following a 4-week HFD is dependent on intrinsic alteration rather than niche perturbation.

- 3) Why does the high fat diet (42%) induce loss of primitive HSCs, whereas the high cholesterol diets increase the HSC population?

In papers from Van den Berg et al. (45% kcal from fat during 18 weeks) and Nagareddy et al. (KO mice models), high fat or high cholesterol diets induce an increase in the hematopoietic stem and progenitor cells (HSPC, also called LSK population). This population contains mostly progenitors and also HSC that are more rarely represented. Our data also confirmed an increase expansion of the HSPC population (LSK) (see Figure 3). Furthermore, in studies from Van den Berg et al. and Nagareddy et al., the authors were working with obese mice, while our study show more that HFD can rapidly increase the LSK population, without being dependent to obesity. Therefore in our paper we have added more discussion about the corroborative data on expansion of HSPC (LSK and MPP) also observed by Van den Berg et al. and Nagareddy et al.

In addition, we analyzed another primitive population by using markers in flow cytometry, and we used CD34 to gate on more primitive HSC (LSK-CD34). It is specifically on this primitive population of HSC that we have shown exhaustion and loss of primitive HSC (see Figure 3). Loss of the primitive HSC (LSK-CD34) has never been described in published work.

4) Nagareddy et al (Cell metabolism 2014) shows different results than described in this paper, please discuss and explain the differences).

By using different obese murine models, Nagareddy et al. described how adipose tissue macrophages can promote proliferation of bone marrow myeloid progenitors. We have now added this important paper to the second paragraph of the discussion. We further explain and discuss the difference observed in our work which was, essentially, that our model was not characterized by obesity and there was no alteration in the bone marrow niche.

To confirm the non-implication of the bone marrow niche in loss of HSC, we studied transplantation (Supplementary Figure S4d-f). We also performed an additional experiment showing that short-term feeding with HFD did not alter the BM structure, which maintained the same number of fat cells (adipocytes) and there was no change in their size either (new Supplementary Figure S4a-c).

5) What other lipid raft dependent pathways are disturbed? Only TGFbeta?

For this important question, we assessed the localization of several other receptors (such as c-Kit, IL3R α and IL6R α) also known to be positioned within lipid rafts. Following HFD, we did not observe significant changes on the clusterization of such receptors (new Supplementary Fig.8). These experiments suggest that HFD perturbs merely Tgfbr1 localization and the downstream phosphorylation of Smad2/3, a canonical effector in TGF- β signaling. Furthermore, injecting mice with recombinant TGF- β 1 compensated the loss of HSC and reactivated TGF- β signaling (Fig. 7 and new Supplementary Fig.S10).

6) What are the results of the lipid raft disturbances, loss of Caveolin dependent endocytosis?

Yes, we agree with the reviewer, it is important to determine the potential results of this lipid raft disturbance. We attempted to address this question by performing immunofluorescence microscopy-based imaging of Caveolin-1 and LR staining in LSK-CD34⁻ LR^{hi} cells from CD and HFD-med mice and we observed that caveolin-1 was strongly co-localized with LR staining taken from HFD-fed mice (see Figure below). So, loss of caveolin-dependent endocytosis could be an option, but to confirm this, it would be necessary to administrate an inhibitor of caveolin endocytosis, which is an experiment impossible to conduct *in vivo* on our mice for ethical reasons. The ethics committee from our University refused such an experiment because such inhibitor is likely to be toxic *in vivo*. Accordingly, we decided not to include the Figure below within the scope of the new version of manuscript.

Figure: Colocalization of CTB-stained LR and the caveolin-1 in LSK-CD34⁻ LR^{hi}. Immunofluorescence is performed by using antibody against Caveolin-1 (Sigma Aldrich) and secondary anti-mouse AF568 antibody (Thermo Fisher Scientific) and AF647-conjugated CTB (Thermo Fisher Scientific).

Reviewer #3's comments:

Hermetet et al have asked whether high-fat diet influences the function of hematopoietic stem cells (HSC). The investigators studied lipid rafts in HSC and found that the most primitive HSC contain high levels of lipid rafts and that these HSC have better reconstitution capacity after bone marrow transplantation than HSC with lower levels of lipid rafts. The authors gave the mice high fat diet and found that this diet after only 4 weeks induced a loss in HSC with high levels of lipid rafts. Furthermore, the high fat diet induced reduced HSC function as measured by competitive transplantation. HSC from treated mice exhibit a lower fraction of quiescent HSC and target genes of TGF-beta showed reduced transcription (p21, p57, etc.). The distribution of lipid rafts in HSC was altered on high-fat treated HSC and the distribution of TGF-beta receptors within lipid rafts was altered as well. When high-fat treated mice were given TGF-beta1, loss of HSC and reduced reconstitution ability was prevented, suggesting that the high-fat diet induces alterations in lipid rafts that reduce or prevent the action of TGF-beta within HSC, thereby inducing a loss in quiescent HSC.

The experiments in this paper are well performed, the approach is thorough and the conclusions are sound. This represents original and innovative research where the authors also provide a mechanism for the findings. I have only minor criticism and suggestions as follows:

We would like to acknowledge the reviewer for the constructive review on our manuscript. To help track changes, all modifications have been marked in red in the main text as well as in the supplementary data and in figure legends.

- 1) Page 4, paragraph about TGF-beta. ...homeostasis of HSC from HFD treated mice LSK-CD34-... The authors should clarify that these HSC are from HFD treated mice.

Yes, we have clarified this now.

- 2) Page 4 next paragraph.as LR had more diffused distribution on CD-HSC... The word distribution is missing.

Yes, we added this word.

- 3) Same paragraph third line from bottom "change" instead of "changed".

Yes, we corrected this.

- 4) Page 4, last paragraph. "In a way to rescue the phenotype" must be meant to say "In an attempt to rescue the phenotype".

Yes, we corrected this.

- 5) In the discussion it would be interesting to discuss why the authors think that the effect is so specific for HSC. TGF-beta acts powerfully on hematopoietic progenitors, not the least erythroid progenitors. Why is there no effect on the progenitors?

Moreover, a HFD led to loss of LR^{hi} cells among HSC (LSK-CD34⁺), whereas LR^{hi} cells among more mature progenitors (such as LK and Lin⁺) were not affected. We observed that LSK-CD34⁺ cells expressed more the Tgfr1 (see Supplementary Fig.S7 in the revised manuscript). Consequently, we can suggest that LR/TGF- β signaling is highly sensitive to HFD intake. This is mentioned at the end of the Discussion.

- 6) It would be interesting if the authors comment in the discussion on the dose of the high-fat diet. Is this dose much higher than humans will normally eat if they eat a lot of fat or is it similar? Are high-fat eaters likely to get reduced HSC function?

Yes, it is interesting point that we decided to introduce in the Discussion section of the manuscript with the following sentence; "Studies on the effects of a HFD on the homeostasis of human HSC and progenitors is rather difficult to conduct, and rodent models are therefore often used. The Western diet contains 42% fat and is therefore comparable to "fast food" which contains an estimated 35-45% fat."

REVIEWERS' COMMENTS:

Reviewer #1 (Remarks to the Author):

I am satisfied with the authors' revision.

Reviewer #2 (Remarks to the Author):

The manuscript has improved significantly, well done! No further comments.

Reviewer #3 (Remarks to the Author):

No comments.

REVIEWERS' COMMENTS ON REVISED MANUSCRIPT:

Reviewer #1 (Remarks to the Author):

I am satisfied with the authors' revision.

We would like to acknowledge the reviewer for his/her careful review and relevant comments, which helped us to improve the quality of our manuscript and to highlight the impact of our work.

Reviewer #2 (Remarks to the Author):

The manuscript has improved significantly, well done! No further comments.

We would like to acknowledge the reviewer for his/her careful review and relevant comments, which helped us to improve the quality of our manuscript and to highlight the impact of our work.

Reviewer #3 (Remarks to the Author):

No comments.

We would like to acknowledge the reviewer for his/her careful review and relevant comments, which helped us to improve the quality of our manuscript and to highlight the impact of our work.